# How is cervical cancer screening information communicated in UK websites? Cross-sectional analysis of content and quantitative presentation formats

Yasmina Okan [ORCID],[1] Samuel G Smith,[2] Wändi Bruine de Bruin [ORCID] [1,3]

[1]Centre for Decision Research, Leeds University Business School, University of Leeds, Leeds, UK
[2]Leeds Institute of Health Sciences, University of Leeds, Leeds, UK
[3]Department of Engineering and Public Policy, Carnegie Mellon University, Pittsburgh, Pennsylvania, USA

**Correspondence to**
Dr Yasmina Okan;
y.okan@leeds.ac.uk

## ABSTRACT

**Objectives** To investigate whether UK websites about cervical cancer screening targeted to the public include (1) information about benefits and risks of screening, possible screening results and cervical cancer statistics, (2) quantitative presentation formats recommended in the risk communication literature and (3) appeals for participation and/or informed decision-making.

**Design** Cross-sectional analysis of websites using a comprehensive checklist of information items on screening benefits, risks, possible results and cervical cancer statistics.

**Outcome measures** We recorded the number of websites that contained each of the information items, and the presentation format used for probabilistic information (no quantification provided, verbal quantifiers only, different types of numerical formats and/or graphs). We also recorded the number of websites containing appeals for participation and/or informed decision-making.

**Setting** Websites were identified through the most common Google search terms used in the UK to find information on cervical screening, according to GoogleTrends and a commercial internet-monitoring programme. Two additional websites were identified by the authors as relevant.

**Results** After applying exclusion criteria, 14 websites were evaluated, including websites of public and private health service providers, charities, a medical society and a pharmacy. The websites mentioned different benefits, risks of screening and possible results. However, specific content varied between websites. Probabilistic information was often presented using non-recommended formats, including relative risk reductions to express screening benefits, and verbal quantifiers without numbers to express risks. Appeals for participation were present in most websites, with almost half also mentioning informed decision-making.

**Conclusions** UK websites about cervical cancer screening were generally balanced. However, benefits and risks were presented using different formats, potentially hindering comparisons. Additionally, recommendations from the literature to facilitate understanding of quantitative information and facilitate informed decisions were often not followed. Designing websites that adhere to existing recommendations may support informed screening uptake.

### Strengths and limitations of this study

► We analysed the content of UK websites about cervical cancer screening using an established checklist of information items, and identified additional information that was mentioned commonly.

► We systematically examined whether cancer screening websites present probabilistic information in formats recommended in the risk communication literature.

► We identified websites by applying the most common Google search terms used in the UK to find information on cervical screening, and examined the majority of links that users will realistically access.

► All the information items we assessed may not be relevant for screening decisions, whereas we may have omitted some information that can be relevant (eg, about the human papillomavirus).

► We did not examine whether websites mentioned the uncertainty associated with estimates of benefits and risks, although such information can be important for informed decision-making.

## INTRODUCTION

Cervical cancer is highly preventable. It is caused in most cases by an infection with the human papillomavirus (HPV), which may lead to abnormal changes in cervical cells.[1 2] Such abnormalities can be detected through screening and treated before they become cancerous.[3] Indeed, cervical screening significantly reduces both cervical cancer incidence and cancer-specific mortality.[3–7] In the UK, age-standardised incidence was 11.8 in 100 000 in 2014, and age-standardised mortality was 3.3 in 100 000.[8] In England alone, estimates suggest that there would be 1827 additional cervical cancer deaths per year without screening.[7]

However, cervical screening is associated with potential risks, including the detection of indolent abnormal cells that would have

cleared up on their own (ie, overdiagnosis), potentially leading to unnecessary treatment.[9 10] A recent meta-analysis indicates that about half of untreated moderately abnormal cells regress within 2 years,[9] suggesting that overtreatment is relatively common. The removal of large amounts of tissue during treatment can also increase the risk of preterm birth in future pregnancies.[11–13] However, experts and policy-makers generally agree that benefits of cervical screening outweigh potential risks.[5 14 15] In the UK, the National Health Service (NHS) offers screening to all women aged 25–64. It has been emphasised that screening invitees need information about both benefits and risks to make informed decisions about participation.[14 16 17]

Organised screening programmes often use written outreach materials. In the UK, eligible women receive an invitation letter with an information leaflet that mentions websites about cervical screening. Hence, it is important to examine whether widely accessed websites effectively support decisions about screening participation. Both European and UK guidelines have emphasised that cancer screening communications should be comprehensive in content, and should provide balanced discussions of benefits and risks to screening.[17–20] Such guidelines have also highlighted that screening communications should be comprehensible, and avoid quantitative presentations that are hard to understand.[17–19] Communications that are not well understood can cause undue concern, reduce recipients' self-efficacy beliefs about their capacity to participate in screening and undermine informed decision-making.[21 22]

Quantitative information in screening communications, however, can be challenging even for educated audiences.[21 23] Such information can be presented using a range of formats, including verbal quantifiers (eg, 'low' or 'moderate' risk), various numerical formats and graphical displays (eg, icon arrays). Research in risk communication has shown that quantitative presentation formats vary considerably in their effectiveness to promote accurate understanding. Some formats are known to be relatively ineffective, while others still lack conclusive evidence.[24] Table 1 provides an overview of quantitative formats, associated evidence-based recommendations and supporting references. Key recommendations relevant to screening communications include avoiding the use of verbal quantifiers without accompanying numbers, numerical '1-in-X' formats and presentations of risk reductions in relative terms. Another recommendation is to add simple graphical displays to numerical information.

Existing guidelines also state that screening communications should not persuade people to attend or present screening as necessary or important, without acknowledging that not participating is a reasonable choice.[17–20] Recent guidance from the UK National Screening Committee[17] notes that 'information should make it clear that it is a personal choice to accept or decline screening and both choices will be fully supported' (p6). This approach to screening communications seeks to respect

**Table 1** Evidence-based recommendations from the risk communication literature to promote understanding of probabilistic information

| Recommendation | Rationale | Key references |
|---|---|---|
| Avoid the use of verbal quantifiers without numbers (eg, women who have abnormal cells removed are *slightly more likely* to have their baby early). | Interpretations of verbal quantifiers vary across individuals and often differ from interpretations intended by communicators. Verbal quantifiers can lead to overestimations of risks. | Budescu et al[77]; Knapp et al[55 56]; Peters et al[57]; Visschers et al[53]; Young and Oppenheimer[58]; Zipkin et al[38] |
| Avoid numerical '1-in-X' formats to present the chance of an outcome (eg, 1 in 12 women will have an abnormal test result). | People tend to perceive the same probabilities as higher and more worrying when presented using '1-in-X' ratios, as compared with numerically equivalent 'N-in-X*N' ratios (eg, 10 in 120 women will have an abnormal test result) or percentages (eg, 8% of women will have an abnormal test result). | Pighin et al[78 79]; Sirota et al[54 80 81]; Trevena et al[48]; Zikmund-Fisher[39] |
| Avoid presenting estimates of risk reduction in relative terms (eg, screening cuts the risk of getting cervical cancer by 75%). | Relative risk differences can obscure the true magnitude of benefit or harm, as compared with absolute risk differences (eg, screening reduces the risk of getting cervical cancer from 20 in 1000 to 5 in 1000). | Akl et al[82]; Covey[83]; Fagerlin et al[22]; Gigerenzer et al[21 84]; Trevena et al[48]; Zipkin et al[38] |
| Add simple graphical displays of numerical information (eg, icon arrays, where icons of different colours represent those affected and not affected by the risk). | Well-designed, simple graphs help to overcome difficulties in understanding numerical information and are often perceived as more appealing and easier to understand. | Galesic et al[85]; Garcia-Retamero and Cokely[40]; Gigerenzer and Edwards[84]; Okan et al[86]; Paling[87]; Schapira et al[88] |

personal autonomy and ensure that decisions are in line with invitees' personal values and circumstances—especially since healthy individuals can be adversely affected by screening.[25 26] Besides undermining the principle of autonomy, persuasive messages may have unintended negative effects, such as eliciting guilt and anxiety among invitees who decline the offer, anger among those who participate and are harmed as a result, and potential mistrust in communicators over time.[20 26 27]

We investigated whether UK websites about cervical screening adhere to existing guidelines and policy recommendations concerning information content and appeals for informed decision-making (vs for participation), as well as to recommendations from the risk communication literature concerning quantitative presentation formats. Previous website analyses have primarily focused on breast cancer screening, and have generally assessed only website content[25 28] or a specific aspect of presentation format (eg, consistency in the presentation of statistics on overdiagnosis[29]).

We used an established checklist of information items about cervical screening,[30] building on earlier evaluations of breast cancer screening communications.[25 31 32] We assessed quantitative presentation formats considering the recommendations listed in table 1. Additionally, we examined the type of appeals included in websites. In sum, our research questions were:

1. Do UK websites about cervical screening contain (a) key information about screening benefits, risks, possible results, and (b) cervical cancer statistics?
2. Do UK websites about cervical screening present probabilistic information using recommended formats?
3. Do UK websites about cervical screening contain appeals for participation and/or informed decision making?

## METHODS
### Search strategy
We used GoogleTrends to identify the most common Google search terms used in the UK to find information on cervical screening. We identified search terms related to 'cervical screening' in the 12 months prior to March 2017. We used 'cervical screening' based on the invitations from the UK's NHS. Related search terms included 'cervical cancer screening,' 'cervical smear' and 'smear test'. To determine which of these terms was more common, we used Wordtracker (https://app.wordtracker.com/), a commercial programme that estimates the relative frequency of Google search terms in a given period of time (see Ref. 33 for a similar procedure). The two most commonly used terms in the UK in the year before March 2017 were 'smear test' and 'cervical screening'.

### Website selection
On 9 March 2017, we performed the Google search 'smear test' OR 'cervical screening' using the private Firefox browsing mode to prevent previously visited pages

and cookies from influencing search results. Following Ghanouni et al,[29] we examined the first five pages of results using the default of 10 results per page, (i.e., 50 links). This includes the vast majority of websites that users will access (see also Ref. 34). The full list of links is available at the Open Science Framework (https://doi.org/10.17605/OSF.IO/73GFN).

We excluded links that (1) targeted healthcare professionals and academics rather than laypeople (eg, technical reports, research articles), (2) reflected online media articles or press releases, (3) presented international information not applicable to the UK (eg, Wikipedia's overview of screening recommendations in different countries) and (4) were locally or regionally oriented. The latter links typically focused on basic practical aspects (eg, who is eligible for screening and logistics of making appointments for specific general practice surgeries), and often provided links for further information to national websites included in our analyses. We also excluded links that (5) were duplicate, (6) contained no or little information about cervical screening (<150 words) and (7) had no written materials (eg, YouTube video). Additionally, we included two websites identified through our knowledge of resources on cervical screening (Patient and Women's Health Concern). Although these websites did not appear in the Google search, they represent trusted UK resources that some women may access directly to learn about cervical screening. Figure 1 summarises the website selection process.

For the websites identified through Google, we coded the link listed in the search results. For the two additional websites, we coded the main link providing information about cervical screening. In all cases, we also coded sections on cervical screening within each website that could be directly accessed from the initial link, as well as other written materials directly accessible, including electronic leaflets, fact sheets and slideshows, which were all considered as part of the same resource in analyses. Websites were accessed between 20 March 2017 and 28 April 2017, except one website (BootsWebMD), which was accessed at a later date due to an oversight in the initial website selection. The date of accessing each website, corresponding links and estimated number of visits appears in supplementary materials (online supplementary table S1). Copies of PDF files reflecting the content of all websites at the time of access are available at the Open Science Framework.

### Coding of websites
To code website content, we adapted a checklist of information items that was developed to analyse invitations for cervical screening.[30] Following European and UK guidelines,[18 19] the checklist included items about screening benefits, risks and possible results (eg, the possibility of an abnormal or an inadequate result). Following Kolthoff et al,[30] the item on overdiagnosis/overtreatment included any reference to screening possibly detecting abnormal cells that may clear up on their own and/or leading to unnecessary treatment, regardless of

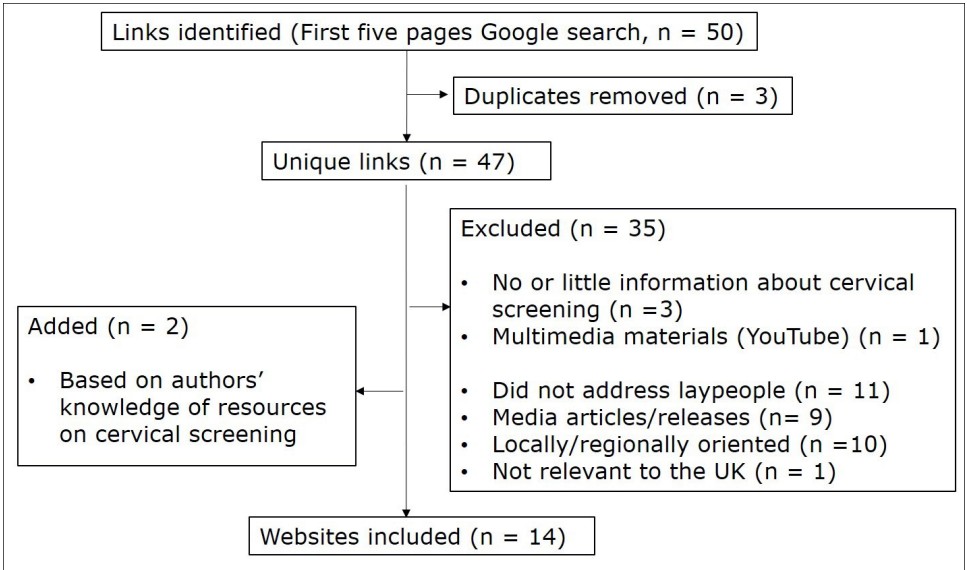

**Figure 1** Flow chart representing the website selection process.

whether the treatment type was mentioned (see also Refs).[35 36] The checklist also included items about cervical cancer statistics (eg, lifetime risk of developing cervical cancer), which may be relevant for screening decisions.[25 31 32] After initial inspection of the websites, we added five items to the checklist to reflect commonly presented information that may also influence screening decisions (figure 2). Examples of all information items appear in supplementary materials (online supplementary table S2).

To assess quantitative presentation formats, we coded whether probabilistic information was presented using: no quantifiers, verbal quantifiers only, numerical quantifiers (1-in-X vs other numerical format) and/or graphs. We also coded whether any information about risk reduction was presented in relative terms. Multiple formats (eg, graphs accompanied by numbers) were present in some communication materials (online supplementary table S3, supplementary materials).

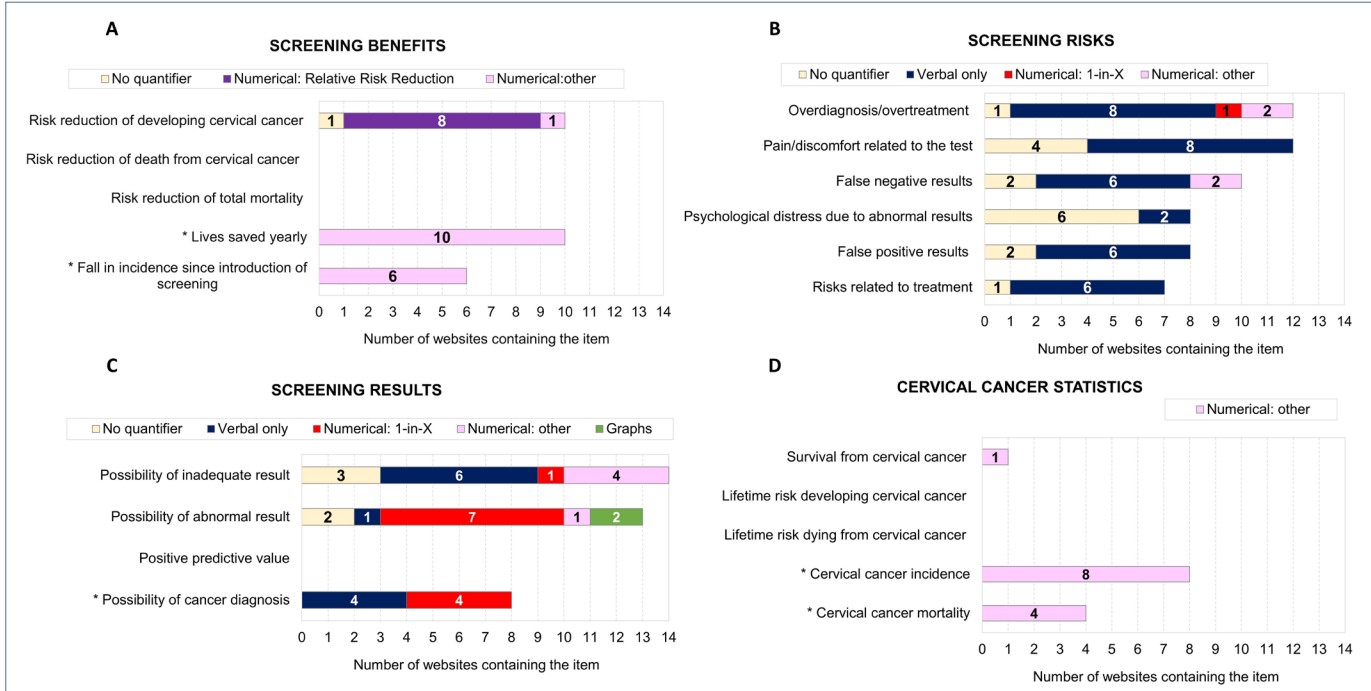

**Figure 2** Results of analysis of content and quantitative presentation formats for information about (A) cervical screening benefits, (B) risks, (C) possible results and (D) cervical cancer statistics. A total of 14 websites were analysed. Items marked with an asterisk were added by the authors to the checklist by Kolthoff *et al.*[30] Details concerning instances in which multiple formats were used for a given information item appear in supplementary materials (online supplementary table S3).

Finally, we coded appeals for informed decision-making (eg, 'deciding whether to have screening or not is your choice') and appeals for participation. Following Kolthoff et al,[30] the latter included direct encouragement for participation (eg, 'a smear test—have it') and statements presenting screening as necessary (eg, 'it's essential for women to have regular cervical screening tests') or important (eg, 'it is important to go for screening every time you are invited').

The first author (YO) read through all selected materials and applied the coding scheme. Three websites (>20%) were coded by another author (SGS). Cohen's kappa was moderate to high on all codes except 'type of appeal' (0.18). Discrepancies for this code were caused by appeals for participation that were not prominent. These discrepancies were resolved through discussion, resulting in consensus. The average Cohen's kappa was 0.86 (range: 0.65–1).

### Patient and public involvement

Patients and/or public were not involved in the design or conduct of the study, which did not involve human participants.

## RESULTS
### Website selection

As described above, we identified 50 links through our Google search. After exclusions, 12 websites remained, including 4 websites from public health service providers in the UK's four countries (NHS Choices England, NHS Inform Scotland, Public Health Agency Northern Ireland, Public Health Wales), 1 private health service provider (Bupa UK), 3 cancer charities (Cancer Research UK, Jo's Cervical Cancer Trust, MacMillan Cancer Support), 2 additional charities (LGBT Foundation, Marie Stopes UK), 1 medical society (British Society for Colposcopy and Cervical Pathology) and 1 pharmacy (Boots WebMD). All websites from public health service providers included links to official invitation leaflets, which were coded with the corresponding website, as described above. As noted earlier, two additional websites (Patient and Women's Health Concern) were identified by the authors. Below, we present summary statistics across websites. Results for individual websites appear in supplementary materials (online supplementary table S3).

### Do UK websites about cervical screening contain (a) key information about screening benefits, risks, possible results and (b) cervical cancer statistics?

We report on four main findings regarding website content. First, screening benefits were mentioned frequently, although the focus was primarily on risk reduction of developing cervical cancer (figure 2A). Estimates of risk reduction typically varied between 70% and 80%. Based on Sasieni et al,[37] two websites provided an estimated range of 60%–80%, and another provided an estimate of 90% for the reduction in the risk of advanced

cancer specifically. No websites explicitly mentioned risk reductions of death from cervical cancer or total mortality. Instead, benefits were often expressed in terms of the number of lives saved yearly (eg, 'cervical screening saves 5000 lives a year in the UK'; n=10 websites), or in some cases referring to the fall in cervical cancer cases since the national screening programme was initiated (eg, 'since the screening programme was introduced in the 1980s, the number of cervical cancer cases has decreased by about 7% each year'; n=6).

Second, screening risks were also mentioned relatively frequently (figure 2B), but the specific risks and their descriptions varied across websites. The most commonly mentioned risks were overdiagnosis and overtreatment (n=12), pain/discomfort related to the test (n=12) and the possibility of false negatives (n=10). Descriptions of overdiagnosis, however, generally focused on cell changes often clearing up and not progressing into cancer (eg, 'in most cases, the abnormal cells will disappear on their own'). The risk of unnecessary additional tests or treatment was only mentioned explicitly in seven cases, often in connection to justifications for current screening age ranges (eg, 'changes in a young woman's cervix are quite normal. In this situation, screening may lead to unnecessary treatment'). Most websites described the test as uncomfortable but not painful (eg, 'having the speculum put in may be a little uncomfortable, but it shouldn't hurt'), though one website did state that the test was potentially painful. Risks related to treatment were only mentioned in half of the websites, all of which referred to the possibility of premature birth. Two of them also noted the risk of stenosis (ie, the cervix becoming tightly closed after treatment).

Third, possible screening results were mentioned in most websites (figure 2C). Estimates of the likelihood of inadequate results varied between 1.7% and 3% and estimates of the likelihood of abnormal results ranged between 5% and 10%. Additionally, the possibility of cancer diagnosis was mentioned in over half of the websites (n=8) and was often reported as occurring in less than 1 in 1000 cases.

Finally, cervical cancer statistics such as lifetime risk of developing or dying from cervical cancer were not discussed, although some websites did provide details on cervical cancer incidence and mortality (figure 2D). Specific estimates of incidence varied across websites, and included yearly, daily, national and regional estimates (eg, 'each year in Northern Ireland, about 103 women are diagnosed with cervical cancer').

### Do UK websites about cervical screening present probabilistic information using recommended formats?

Information was often not presented in recommended formats. First, the recommendation to avoid the use of verbal quantifiers without numbers[38] was often not followed for information about risks (figure 2B). Risks related to treatment were quantified only verbally, generally using varying verbal quantifiers. For example, one

website stated that 'women are *slightly more likely* to have their baby 1 to 2 months early', and another stated that 'women may have a *higher risk* of preterm delivery'. Other risks were also either only quantified verbally or not quantified at all, with the exception of overdiagnosis/overtreatment (n=3) and false-negative results (n=2).

Second, the recommendation to avoid '1-in-X' numerical formats[39] was often violated for information about screening results. This was the most popular format for conveying the likelihood of abnormal results (figure 2C). Third, the recommendation to avoid presentations of relative risk reduction[21] was violated in most cases in which an estimate of risk reduction was provided (figure 2A). Finally, the recommendation to add simple graphical displays[40] was often not met. Only two websites contained graphs, each depicting the likelihood of abnormal results.

### Do UK websites about cervical screening contain appeals for participation and/or informed decision-making?

Appeals for participation were present in most websites (n=12). Specifically, appeals for participation without mention of informed decision-making occurred in half of the websites (n=7). Several websites combined appeals for participation and informed decision-making (n=5) (eg, 'don't ignore your smear test, it could save your life' and 'taking part in cervical screening is your choice'). One website (NHS Choices England) referred to informed decision-making only ('deciding whether or not to have a cervical screening test is your choice'). The websites of the three remaining public health service providers included either both appeals for participation and informed decision-making (NHS Inform Scotland and Public Health Wales) or only appeals for participation (Public Health Agency Northern Ireland).

### DISCUSSION

We investigated whether UK websites about cervical screening adhered to existing recommendations about content, quantitative presentation formats and type of appeals. We found that websites often followed recommendations from European and UK guidelines to include information about both benefits and risks of screening.[17–19] However, specific content varied across websites. For example, only half of the websites explicitly referred to risks related to the treatment of abnormal cells. Additionally, probabilistic information was often presented in formats not recommended in the risk communication literature, including relative risk reductions to express screening benefits, and verbal quantifiers without numbers to express risks. Moreover, several websites encouraged screening participation without referring to informed decision-making, contrasting with current UK policy.[14 41]

Our finding that websites often included information about both benefits and risks of cervical screening contrasts with previous reports that breast screening websites lacked information about risks such as overdiagnosis.[25 28] Recent emphasis in the UK on facilitating informed screening decisions[14 41] may have led to more comprehensive and balanced websites in recent years (see also Ref. 29). The presence of information about both screening benefits and risks is aligned with women's preferences, who often want to receive information on both aspects before their test.[42] However, we also found that websites generally presented benefits and risks using different formats (numbers vs verbal quantifiers, respectively), potentially hindering benefit–risk comparisons. Additionally, our finding that specific content varied across websites suggests that women accessing different resources may come to different conclusions.

Our results also suggest that existing descriptions of benefits and risks may not always support understanding. For example, the concepts of overdiagnosis and overtreatment are often unfamiliar and counterintuitive to people,[43–45] highlighting the need to provide explanations that are accessible and transparent. Many websites did explain that cell changes often clear up on their own. This may help women to understand that cervical screening can lead to overdiagnosis of indolent abnormalities (potentially causing unnecessary anxiety or worries[36]), but not necessarily that it may result in unnecessary treatment. Moreover, the websites that explicitly mentioned overtreatment often did so in relation to women below the recommended screening age. Hence, those within the recommended age may incorrectly infer that such risk does not apply to them. We also found that many websites emphasised that screening 'saves lives'. It has been argued that such claims are misleading, and that communications should report reductions in cancer-specific mortality, overall mortality, as well as overall cancer deaths.[46] Although there is evidence that cervical screening reduces cancer-specific mortality,[3–7] determining its impact on overall mortality requires large trials with sufficient power to detect differences.[47]

Our finding that probabilistic information was often presented in non-recommended formats is concerning as this may cause misperceptions, even among educated audiences. The use of relative risk reductions to express screening benefits can hinder understanding and increase risk perceptions, relative to presentations of absolute risk reduction.[38 48] Although information about baseline risk (ie, the absolute risk without screening) could reduce the biasing effect of relative risks[49] (but see Ref. 50), such beneficial impact may be limited to baseline risks presented in frequency formats (vs probabilities)[51 52] and to more (vs less) numerate recipients.[52] Moreover, we did not find such information in any of the websites. It has been argued, however, that presentations of relative risks may be considered for low-probability risks with high impact (eg, an earthquake), to prevent people from neglecting such risks altogether.[24 53]

Additionally, our results suggest that women may overestimate the likelihood of abnormal screening results due to the use of '1-in-X' formats (eg, '1 in 12 women will have an abnormal test result').[54] Similarly, screening risks

expressed using verbal quantifiers without numbers may be overestimated.[55–58] The absence of numerical information can also lead people to perceive communications as less credible and trustworthy.[53 59 60] Although some risks are hard to quantify (eg, psychological distress), estimates are available for risks such as preterm birth or overtreatment, from observational studies and meta-analyses.[9 11 13] The best available evidence could be presented in transparent fact boxes—a tabular format that facilitates comparisons of outcomes in groups of screened versus unscreened individuals.[61] Fact boxes can also include simple graphs such as icon arrays to allow visual comparison of quantities,[62] which could be especially beneficial for people with low numeracy. Yet, graphs were seldom used in cervical screening websites.

Finally, our finding that some websites encouraged screening without mentioning informed decision-making contrasts with current UK policy, which emphasises that communications can recommend screening, but should acknowledge that not accepting the offer is a reasonable choice.[14 20 41] Although there is largely agreement that benefits of cervical screening outweigh potential risks,[5 14 15] persuasive messages raise ethical concerns to the extent that harms are possible.[18 26 63] Persuasive messages may also contribute to widespread enthusiasm for cancer screening,[64 65] discouraging people from reasoning about their screening choices.[26] Moreover, while guidelines for screening communications tend to focus on health service providers,[17–19] conflicting recommendations from different sources might create public confusion, negative beliefs about recommendations and scientific research, and potentially reduce screening intentions.[66 67]

Strengths of our analyses include their comprehensiveness and systematic examination of website content using an established checklist of information items,[30] and recommendations from the literature for quantitative presentation formats. Additionally, our analyses included most of the websites that UK citizens may realistically access, as our strategy for website selection was based on the top results identified by the most commonly used Google search terms. Our approach could be used by researchers and practitioners to evaluate the content and format of websites about related topics (eg, other types of screening) in different countries. This would allow comparisons of websites for countries that have organised versus opportunistic screening. For example, website content may be more heterogeneous in countries such as the USA, which do not have organised screening. It would also be interesting to compare different platforms (eg, desktop websites vs mobile tailored versions), and to examine whether different design features (eg, information position[68]) affect information readability, saliency or users' information seeking behaviour .

Limitations of our study include that our analyses may not have covered all information that is potentially relevant for screening decisions. For example, we did not assess whether websites discussed HPV, or the uncertainty associated with estimated benefits and risks.

Communicating uncertainty can be challenging,[69–71] but is important for informed decision-making.[16 17 41] Conversely, it is possible that not all codes assessed are essential for informed screening decisions. We also did not directly examine whether users can make informed screening decisions based on the different websites. It is likely, however, that following guidelines and recommendations from the literature will facilitate better understanding and more informed decisions. Explicit appeals to informed decision-making may also encourage users to evaluate the content, but of course do not guarantee that informed decisions will occur.

Future work should aim to identify essential information items for informed decisions about screening, considering both experts' and women's views. As noted by Ghanouni *et al*,[72] guidelines often do not provide detailed advice on *which* specific screening benefits, risks, or results should be discussed in communications. Moreover, experts may not fully agree on the relative importance of different information items, and women may have different information needs and preferences, including about whether and when to receive information about further tests. Providing too much unfamiliar information initially may also overwhelm screening invitees, and potentially distract them from key information necessary for decision-making.[22 73] More research is also needed to identify how to best convey the individual significance of the population-level estimates included in screening communications, which may be hard for people to understand.[74] Finally, future studies should also examine how screening decisions (eg, intentions to participate[75]) are affected by information about specific benefits and/or risks presented using different formats, building on the risk communication literature (eg,[76]). Such work would provide valuable insights to inform the design of evidence-based public communications about screening.

**Acknowledgements** The authors are grateful to Mirta Galesic and Erika Waters for their valuable comments and suggestions. They would also like to thank the reviewers (Jo Waller, Felix G. Rebitschek, and Seok Won Jin) for their constructive feedback.

**Contributors** YO conceived the research, acquired and analysed the data, and drafted the initial version of the manuscript. SGS assisted with data analyses and interpretation. WBdB contributed to data interpretation. All authors contributed to study design, revised the manuscript critically and approved the final version.

**Funding** This work was supported by a Population Research Fellowship awarded by Cancer Research UK to Yasmina Okan (reference C57775/A22182). SS is supported by Yorkshire Cancer Research. WBdB was partially supported by a grant from the Swedish Riksbanken Jubilieumsfond programme on Science and Proven Experience.

**Disclaimer** The funding agencies had no involvement in designing the study, data collection, analysis and interpretation, writing the report, or the decision to submit the article for publication.

**Competing interests** SGS is an academic consultant for Luto, who are not involved in any of the websites reviewed in this manuscript.

**Patient consent for publication** Not required.

**Provenance and peer review** Not commissioned; externally peer reviewed.

**ORCID iDs**
Yasmina Okan http://orcid.org/0000-0001-7963-1363
Wändi Bruine de Bruin http://orcid.org/0000-0002-1601-789X

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
