## [Reviewer comments · BMJ Open]

ARTICLE DETAILS

TITLE (PROVISIONAL)	How is cervical cancer screening information communicated in UK websites? A cross-sectional analysis of content and quantitative presentation formats
AUTHORS	Okan, Yasmina; Smith, Samuel; Bruine de Bruin, Wändi

VERSION 1 - REVIEW

REVIEWER	Jo Waller UCL, UK
REVIEW RETURNED	18-Feb-2019

GENERAL COMMENTS	This is a well-written, clear paper, addressing the important issue of presentation of information about cervical screening. It uses similar methods to previous studies in the breast screening field, but has the added benefit of analysing the way in which probabilistic information is presented, which is of key importance for ensuring good communication. The methods and results are particularly well-described and the figures and tables are really nicely presented and useful. I just have a few small comments for the authors to consider. 1) I would like to have seen a slightly stronger justification of the rationale for including the selected websites. I can see that national websites have wider appeal than regional ones, but if regional ones are coming up near the top of Google searches, might women not be using these? Secondly, the inclusion of the two websites that were known to the researchers isn't self-evident to me. Is there reason to believe that women are using these sites to search for information, even if they are not coming up in the first 5 pages of a Google search? I'm not suggesting changing the websites you've included, but can you give a stronger rationale? 2) There is (quite rightly) much made of guidelines about informed choice, but should all the websites included be subscribing to this approach? I think a possible discussion point is about whether, for example, it might be acceptable for a cervical cancer charity to strongly advocate for screening, with less consideration for personal autonomy, than would be appropriate for Public Health England. I thought the differences between the public health websites of the 4 devolved nations were really interesting (only PHE sticking to informed decision-making) and might be worthy of mention.
---

	3) Finally, I wondered if the authors might consider alluding to the inherent tension between communicating individual-level and population-level risk and benefits of screening. Given that, by definition, only a population (not an individual) can be screened, the risks and benefits have to be calculated at the population level. By encouraging individuals to make informed choices based on their individual chances of benefit and harm, do we risk undermining the whole process? (Because of course, very few individuals will actually benefit from taking part in screening). Maybe it's beyond the scope of this paper, but perhaps something to mention?
--	---

REVIEWER	Felix G. Rebitschek Harding Center for Risk Literacy Max Planck Institute for Human Development Lentzeallee 94 14195 Berlin Germany
REVIEW RETURNED	04-Mar-2019

GENERAL COMMENTS	The manuscript "How is cervical cancer screening information communicated in UK websites?" reviews current UK health information webpages that inform about cancer screening. It shows that substantial variation in information presentation and reveals deficits in promoting informed screening decision-making. This type of paper is of high importance as descriptions of real-world information environments for screening decisions are too sparse. Particularly, since evidence-based medicine has established guidelines of how to present best health information based on risk communication research, the research described here implies corrective measures for public pages. I recommend it for acceptance, after clarifying the following issues:  - Did the authors target information pages on the screening method or about the programmes or mixed – perhaps adapt the title? - How did they select the additional screening outcomes – as benefits and harms are containing patient-relevant clinical study outcomes. What was the rationale for demanding those outcomes that are not defined as screening trial outcomes regularly? If it is due to the used checklist, please check at least against the latest systematic review on cervical cancer screening and explain, what is meant by overdiagnosis/overtreatment: there are punch biopsies and treatments of different types of precancerous lesions. - go more into detail on overdiagnosis/overtreatment in the discussion, as you mention that the concepts are counterintuitive -- need for transparent information – without discussing your findings concerning the concepts - I would appreciate a confirmation showing that laypeople actually land at those pages – with the notion – that you removed international pages (English: US, Canadian, Australian, Northern Europe) – this is really difficult when concluding about the chances of informed decision making. So, here I see a important distinction: your manuscripts reveals an important state of the art in UK health
--

information systems -- however, I am hesitant to believe that laypeople stick with UK pages („most common search teams used in the UK“). As such you could discuss briefly the possibilities of making an informed screening decision based on web search.

- could you please provide current metrics (users, visitors, visits, page impressions or...) for the 14 pages

- how did you deal with written blogs targeting laypeople?

- could you give a brief hint (one sentence in the discussion, outlook), whether are all pages accompanied by an app version of those information (differences across platforms)?

- please highlight that not relative risk presentation per se but without accompanying absolute numbers are problematic

- please clarify under which conditions informed decision making can be seen as supported by a page (mentioning informed decision-making explicitly is not a cue of the relevant quality)

- too strong claim about the second “strength” – there were studies systematically examining cancer screening webpages for recommended risk communication formats (e.g. Hofmann, J., Kien, C., & Gartlehner, G. (2015). Comparative evaluation of information products regarding cancer screening of German-speaking cancer organizations. *Zeitschrift für Evidenz, Fortbildung und Qualität im Gesundheitswesen*, 109(4), 350-362.)

- You revealed something very important: 1) why risk reduction for developing cervical cancer varied dramatically across pages – I think you should highlight wrong numbers, even if presented transparently (with absolute numbers); 2) what is the evidence that cervical cancer screening saves 5000 lives a year in the UK (reference) – Prasad, V., Lenzer, J., & Newman, D. H. (2016). Why cancer screening has never been shown to “save lives”—and what we can do about it. *BMJ*, 352, h6080.

- Minor points

-- infection with HPV is not the only cause of cervical cancer, as one could understand the first sentence

-- easier: the risk of preterm birth due to the removal of abnormal cells

-- please clarify: is the official invitation leaflet included in the 14?

-- if you discuss transparent presentation formats, consider including a glance on methods to build screening information and related effectivity (McDowell, M., Rebitschek, F. G., Gigerenzer, G., & Wegwarth, O. (2016). A simple tool for communicating the benefits and harms of health interventions: a guide for creating a fact box. *MDM policy & practice*, 1(1), 2381468316665365. ; McDowell, M., Gigerenzer, G., Wegwarth, O., & Rebitschek, F. G. (2019). Effect of Tabular and Icon Fact Box Formats on Comprehension of Benefits and Harms of Prostate Cancer Screening: A Randomized Trial. *Medical Decision Making*, 39(1), 41-56.)

	- suggest: "it has been argued that such claims are misleading" - suggest: "it is also possible that not all codes we assessed are essential for..." - please make sure that Figure 2 is larger I would like to read a new version. Felix Rebitschek I always sign my reviews
--	--

REVIEWER	Seok Won Jin The University of Memphis, TN, USA
REVIEW RETURNED	21-Mar-2019

GENERAL COMMENTS	Abstract  • (p. 3, line 15). Design: it would clearer if you could clarify the 'information items,' given the word counts' limit. Introduction  • (p. 5, line 5-8). While most cases of cervical cancer are caused by infection with HPV, not all necessary. Please clarify the statement. • (p. 5, line 5-11). Appropriate in-text citations seem to be needed. • (p. 6, line 21-22). Numeracy is usually known as one of health literacy construct components so it would be clearer to say "health literacy (e.g., numeracy)" Methods  • (p. 9, line 24-39) mechanic error (i.e., "cervical cancer screening"; "smear test", followed by "cervical screening"). The same on p. 14, line 5-9 and 47-48, p. 15 line 19-20, and p. 16, line 3-4. • (p. 9) Website selection: in Introduction, the authors indicated that organized screening programs provide websites about cervical screening (p. 5-6, line 51-6). If so, it is imperative to make sure including these websites for the analyses. Please provide the clarification on this where appropriate in this section.
---

VERSION 1 – AUTHOR RESPONSE

REVIEWER 1 (Jo Waller, UCL, UK)

REVIEWER 1, COMMENT 1: (a) I would like to have seen a slightly stronger justification of the rationale for including the selected websites. I can see that national websites have wider appeal than regional ones, but if regional ones are coming up near the top of Google searches, might women not be using these? (b) Secondly, the inclusion of the two websites that were known to the researchers isn't self-evident to me. Is there reason to believe that women are using these sites to search for information, even if they are not coming up in the first 5 pages of a Google search? I'm not suggesting changing the websites you've included, but can you give a stronger rationale?

AUTHORS' RESPONSE: To address comment 1a, we implemented two actions. First, we clarified the rationale for the exclusion of locally or regionally-oriented websites in the 'Website selection' section

(p. 11): “The latter links typically focused on basic practical aspects (e.g., who is eligible for screening, logistics of making appointments for specific GP surgeries), and often included links for further information to national websites included in our analyses.” Second, we have uploaded a PDF document to the Open Science Framework reflecting the 50 links identified in the Google search, prior to applying the exclusion criteria. As can be seen, the first two pages of Google results (i.e., first 20 links) contained only one locally or regionally-oriented link (i.e., Pathfields Medical Group), with the remaining ones appearing from the third page of results onwards.

To address comment 1b, we have clarified the rationale for the inclusion of the two additional websites (p. 11): “We added two websites identified through our knowledge of resources on cervical screening (Patient and Women’s Health Concern). Although these websites did not appear in the Google search, they represent trusted UK resources that some women may access directly to learn about cervical screening.”

REVIEWER 1, COMMENT 2: There is (quite rightly) much made of guidelines about informed choice, but should all the websites included be subscribing to this approach? I think a possible discussion point is about whether, for example, it might be acceptable for a cervical cancer charity to strongly advocate for screening, with less consideration for personal autonomy, than would be appropriate for Public Health England. I thought the differences between the public health websites of the 4 devolved nations were really interesting (only PHE sticking to informed decision-making) and might be worthy of mention.

AUTHORS’ RESPONSE: We thank the reviewer for raising this issue. We acknowledge that this may be the perspective of some charities. However, we believe that all screening communications should respect patients’ personal autonomy and their right to decide whether or not to undergo any intervention, in line with accepted standards for informed consent (e.g., General Medical Council, 2008). Communications that aim to persuade people to get screened without acknowledging that not doing so is a reasonable choice are at odds with such standards. Moreover, as we now discuss in the manuscript (p. 9): “Besides undermining the principle of autonomy, persuasive messages may have unintended negative effects, such as eliciting guilt and anxiety among invitees who decline the offer, anger among those who participate and are harmed as a result, and potential mistrust in communicators over time[20,48,49].”

In the Discussion (pp. 19-20) we have also elaborated further on this issue: “Finally, our finding that some websites encouraged screening without mentioning informed decision making contrasts with current UK policy, which emphasizes that communications can recommend screening, but should acknowledge that not accepting the offer is a reasonable choice[14,20,60]. Although there is largely agreement that benefits of cervical screening outweigh potential risks[5,14,15], persuasive messages raise ethical concerns to the extent that harms are possible[18,48,75]. Persuasive messages may also contribute to widespread enthusiasm for cancer screening (e.g.,[76,77]), discouraging people from reasoning about their screening choices[48]. Moreover, while guidelines for screening communications tend to focus on health service providers[17–19], conflicting recommendations from different sources might create confusion, negative beliefs about recommendations and scientific research, and potentially reduce screening intentions[78,79].”

As suggested, we now also discuss the differences between the public health websites of the 4 nations within the UK: “One website (NHS Choices England) referred to informed decision making only (“deciding whether or not to have a cervical screening test is your choice”). The websites of the three remaining public health service providers included either both appeals for participation and

informed decision making (NHS Inform Scotland and Public Health Wales) or only appeals for participation (Public Health Agency Northern Ireland).” (p. 17)

REVIEWER 1, COMMENT 3: Finally, I wondered if the authors might consider alluding to the inherent tension between communicating individual-level and population-level risk and benefits of screening. Given that, by definition, only a population (not an individual) can be screened, the risks and benefits have to be calculated at the population level. By encouraging individuals to make informed choices based on their individual chances of benefit and harm, do we risk undermining the whole process? (Because of course, very few individuals will actually benefit from taking part in screening). Maybe it's beyond the scope of this paper, but perhaps something to mention?

AUTHORS' RESPONSE: In the Discussion we now refer explicitly to the challenges associated with communicating population-level estimates, and the need to examine how this impacts individual decisions (p. 21): “More research is also needed to identify how to best convey the individual significance of the population-level estimates included in screening communications, which may be hard for people to understand [86]. Finally, future studies should also examine the impact of different presentation formats on screening decisions (e.g., intentions to participate[87]), as evidence is still scarce.”

Concerning the potential impact of communicating individual-level risks and benefits (e.g., acknowledging that benefits are likely to be small), this is certainly an interesting question, but is indeed beyond the scope of the current paper, which focuses on estimates that can be quantified. As noted by the reviewer, risks and benefits can only be estimated with some precision at the population level.

REVIEWER 2

(Felix G. Rebitschek, Harding Center for Risk Literacy, Max Planck Institute for Human Development)

REVIEWER 2, COMMENT 1: Did the authors target information pages on the screening method or about the programmes or mixed – perhaps adapt the title?

AUTHORS' RESPONSE: The title refers broadly to “cervical cancer screening information” to reflect that we targeted a mix of pages containing information about the programmes and about the screening method, among other aspects. This is also reflected in the broad search terms used (“smear test” and “cervical screening”), described on p. 10. Hence, we believe that it would be best not to modify this aspect of the title.

We should, note, however, that we did revise the title to reflect the study design (see Editorial Office, Request #1). The title now reads: “How is cervical cancer screening information communicated in UK websites? A cross-sectional analysis of content and quantitative presentation formats”

REVIEWER 2, COMMENT 2: How did they select the additional screening outcomes – as benefits and harms are containing patient-relevant clinical study outcomes. What was the rationale for demanding those outcomes that are not defined as screening trial outcomes regularly? If it is due to the used checklist, please check at least against the latest systematic review on cervical cancer screening and explain, what is meant by overdiagnosis/overtreatment: there are punch biopsies and treatments of different types of precancerous lesions.

AUTHORS' RESPONSE: We have addressed this comment in three ways.

First, we have clarified that the additional screening outcomes were based both on the checklist used and on European and UK guidelines, which note that screening communications should also include information about possible screening results: "Following European and UK guidelines[18,19], the checklist included items about screening benefits, risks, and possible results (e.g., the possibility of an abnormal or an inadequate result)." (p. 12)

Second, throughout the paper we have replaced the term "screening outcomes" with "screening results", to clarify that this category focuses specifically on different types of possible results (e.g., abnormal results, inadequate results) and their accuracy (i.e., positive predictive value).

Finally, as suggested we have also clarified what we mean by overdiagnosis/overtreatment (p. 12): "Following Kolthoff et al. [52] the item on overdiagnosis/overtreatment included any reference to screening possibly detecting abnormal cells that may clear up on their own and/or leading to unnecessary treatment, regardless of whether the treatment type was mentioned (see also [57,58])." Of note, both the Introduction and Discussion cite systematic reviews on benefits and risks of cervical screening (e.g., Kyrgiou et al., 2017; Peirson et al., 2013; Tainio et al., 2018)

REVIEWER 2, COMMENT 3: go more into detail on overdiagnosis/overtreatment in the discussion, as you mention that the concepts are counterintuitive -- need for transparent information – without discussing your findings concerning the concepts

AUTHORS' RESPONSE: We have expanded the Discussion (p. 18) to provide more details about our findings concerning overdiagnosis/overtreatment: "For example, the concepts of overdiagnosis and overtreatment are often unfamiliar and counterintuitive to people[62–64], highlighting the need to provide explanations that are accessible and transparent. Many websites did explain that cell changes often clear up on their own, potentially helping women to understand that cervical screening often finds indolent abnormalities (leading to unnecessary anxiety or worries[58]), but not necessarily that it can result in unnecessary treatment. Moreover, the websites that explicitly mentioned overtreatment often did so in relation to women below the recommended screening age. As a result, those within the recommended age may incorrectly infer that such risk does not apply to them."

In the Results section (p. 15) we have also included an additional example of a statement coded under the category of overdiagnosis/overtreatment, to further illustrate how these concepts were described in the websites: "The risk of unnecessary additional tests or treatment was only mentioned explicitly in seven cases, often in connection to justifications for current screening age ranges (e.g., "changes in a young woman's cervix are quite normal. In this situation, screening may lead to unnecessary treatment")."

REVIEWER 2, COMMENT 4: I would appreciate a confirmation showing that laypeople actually land at those pages – with the notion – that you removed international pages (English: US, Canadian, Australian, Northern Europe) – this is really difficult when concluding about the chances of informed decision making. So, here I see a important distinction: your manuscripts reveals an important state of the art in UK health information systems -- however, I am hesitant to belief that laypeople stick with UK pages („most common search teams used in the UK“). As such you could discuss briefly the possibilities of making an informed screening decision based on web search.

AUTHORS' RESPONSE: We have addressed this comment in three ways. First, to show that people actually land at the selected websites, we have added the estimated number of visits globally and percentage of UK visits to the specific websites for which data were available (Supplement, Table S1). As can be seen, estimates of global monthly visits for the different websites range between around 62,000 to over 4 million, with a substantial proportion of estimated UK visitors in most cases. As acknowledged in the footnote for this table, however, estimates are computed by extrapolation from a small panel of users and therefore need to be interpreted with caution (please see also response to comment 5, for further details).

Second, we have uploaded an additional document to the Open Science Framework listing all the 50 links that appeared in the Google search, prior to applying our exclusion criteria. As seen in this document, all 50 links involved UK websites, except one Wikipedia page providing an overview of screening recommendations in different countries (as now noted on p. 11), which was excluded (see also Figure 1 in the manuscript). This illustrates that our decision to exclude international pages had little impact on our results and conclusions. This is not particularly surprising considering that we used the most common search terms used in the UK. Indeed, as noted in the Introduction, our study was designed to examine the content and format of UK websites specifically (and not of all websites accessed by English-speaking users worldwide).

Finally, in the Discussion (p. 20) we now also mention that future work could evaluate websites for different screening programs in different countries: “Our approach could be used by researchers and practitioners to evaluate the content and format of websites about related topics (e.g., other types of screening) in different countries. This would allow comparisons of websites for countries that have organized vs. opportunistic screening. For example, website content may be more heterogeneous in countries such as the United States, which do not have organized screening.”

REVIEWER 2, COMMENT 5: could you please provide current metrics (users, visitors, visits, page impressions or...) for the 14 pages

AUTHORS' RESPONSE: As requested, the Supplement (Table S1) now lists the estimated number of visits globally and percentage of UK visits to websites for which data were available (i.e., 12 out of the 14 websites analyzed). As noted in the footnote to this table, estimates were obtained from SimilarWeb (www.similarweb.com) and represent visits in June 2019. We also note that estimates aggregate data across all website subdomains and are computed by extrapolation from a small panel of users, and therefore should be interpreted with caution.

To the best of our knowledge, it is not possible to obtain direct measurement data for third-party websites. Of note, we also aimed to find estimates for the 2 remaining websites using two additional tools, namely Alexa Internet (www.alexa.com) and Quantcast (www.quantcast.com). However, estimates were not available either.

REVIEWER 2, COMMENT 6: how did you deal with written blogs targeting laypeople?

AUTHORS' RESPONSE: The results of our Google search did not include any blogs targeting laypeople. This can be seen in the new PDF document available at the Open Science Framework, which reflects the 50 links that appeared in the Google search prior to applying our exclusion criteria.

REVIEWER 2, COMMENT 7: could you give a brief hint (one sentence in the discussion, outlook), whether are all pages accompanied by an app version of those information (differences across platforms)?

AUTHORS' RESPONSE: We could not find an app version providing information about cervical screening for any of the websites examined – although there are apps for general health services unrelated to our analyses, such as booking doctor appointments and ordering prescriptions. The majority of the websites that we examined currently did have a mobile-friendly version. However, at this moment we cannot determine whether these versions were available at the time we performed our search. Hence in the Discussion we have added a sentence stating that it would be interesting to examine this in future work: “It would also be interesting to compare different platforms (e.g., desktop websites vs. mobile tailored versions), and to examine whether website design (e.g., information position[80]) optimizes readability.” (p. 20)

REVIEWER 2, COMMENT 8: please highlight that not relative risk presentation per se but without accompanying absolute numbers are problematic

AUTHORS' RESPONSE: In the Discussion (pp. 18-19) we have highlighted that: “Although information about baseline risk (i.e., the absolute risk without screening) could reduce the biasing effect of relative risks[67] (but see [68]), such beneficial effects may be limited to baseline risks presented in frequency formats (vs. probabilities) [69,70] and to more (vs. less) numerate recipients [70]. Moreover, we did not find such information in any of the websites.”

We do also acknowledge that “It has been argued, however, that presentations of relative risks may be considered for low-probability risks with high impact (e.g., an earthquake), to prevent people from neglecting such risks altogether [24,29].”

REVIEWER 2, COMMENT 9: please clarify under which conditions informed decision making can be seen as supported by a page (mentioning informed decision-making explicitly is not a cue of the relevant quality)

AUTHORS' RESPONSE: We fully agree in that mentioning informed decision making explicitly is not a cue of the relevant quality. We have acknowledged this in the Discussion (pp. 20-11), where we now also note that our analysis does not allow concluding whether a given website supports informed

decision making or not: “Limitations of our study include that our analyses may not have covered all information that is potentially relevant for screening decisions. For example, we did not assess whether websites mentioned information about the human papillomavirus (HPV), or the uncertainty associated with estimated benefits and risks. Communicating uncertainty to laypeople can be challenging[81–83], but is important for informed decision making[16,17,60]. Conversely, it is possible that not all codes assessed are essential for informed screening decisions. We also did not directly assess whether users can make informed screening decisions based on the different websites. It is likely, however, that following guidelines and recommendations from the literature will facilitate better understanding and more informed decisions. Explicit appeals to informed decision making may also encourage users to evaluate the content, but of course do not guarantee that informed decisions will occur.”

REVIEWER 2, COMMENT 10: too strong claim about the second “strength” – there were studies systematically examining cancer screening webpages for recommended risk communication formats (e.g. Hofmann, J., Kien, C., & Gartlehner, G. (2015). Comparative evaluation of information products regarding cancer screening of German-speaking cancer organizations. *Zeitschrift für Evidenz, Fortbildung und Qualität im Gesundheitswesen*, 109(4), 350-362.)

AUTHORS’ RESPONSE: Our claim about the second strength has been edited to reflect this, and we no longer state that “this was the first study” to systematically examine presentation formats of probabilistic information.

REVIEWER 2, COMMENT 11: You revealed something very important: 1) why risk reduction for developing cervical cancer varied dramatically across pages – I think you should highlight wrong numbers, even if presented transparently (with absolute numbers); 2) what is the evidence that cervical cancer screening saves 5000 lives a year in the UK (reference) – Prasad, V., Lenzer, J., & Newman, D. H. (2016). Why cancer screening has never been shown to “save lives”—and what we can do about it. *BMJ*, 352, h6080.

AUTHORS’ RESPONSE: To address the first point, we now provide additional details about the risk reduction estimates in the Results section to clarify that the existing variation does not necessarily imply that estimates are incorrect. In doing so we have cited the studies from which estimates were obtained, where applicable: “Estimates of risk reduction typically varied between 70% and 80%. Based on Sasieni et al.[59], two websites provided an estimated range of 60-80%, and another provided an estimate of 90% for the reduction in the risk of advanced cancer specifically.” (p. 14)

To address the second point, we added some of the points by Prasad et al. in relation to documenting impact on overall mortality: “We also found that many websites emphasized that screening “saves lives”. (...) Although there is evidence that cervical screening reduces cancer-specific mortality[3–7], determining its impact on overall mortality requires large trials with sufficient power to detect differences [66].” (p. 18)

- Minor points

REVIEWER 2, COMMENT 11: infection with HPV is not the only cause of cervical cancer, as one could understand the first sentence

AUTHORS' RESPONSE: We have clarified that "it is caused in most cases by an infection with the human papillomavirus (HPV)"

REVIEWER 2, COMMENT 12: easier: the risk of preterm birth due to the removal of abnormal cells

AUTHORS' RESPONSE: We have revised this sentence to also clarify that the risk may depend on the amount of tissue removed: "(...) increasing risk of preterm birth as relatively larger amounts of tissue are removed during treatment" [11–13]"

REVIEWER 2, COMMENT 13: please clarify: is the official invitation leaflet included in the 14?

AUTHORS' RESPONSE: We have clarified this: "After exclusions, 12 websites remained (Figure 1), including 4 websites from public health service providers in the UK's four countries (NHS Choices England, NHS Inform Scotland, Public Health Agency Northern Ireland, Public Health Wales) (...) All websites from public health service providers included links to official invitation leaflets, which were coded with the corresponding website, as described above." (pp. 13-14)

REVIEWER 2, COMMENT 14: if you discuss transparent presentation formats, consider including a glance on methods to build screening information and related effectivity (McDowell, M., Rebitschek, F. G., Gigerenzer, G., & Wegwarth, O. (2016). A simple tool for communicating the benefits and harms of health interventions: a guide for creating a fact box. *MDM policy & practice*, 1(1), 2381468316665365. ; McDowell, M., Gigerenzer, G., Wegwarth, O., & Rebitschek, F. G. (2019). Effect of Tabular and Icon Fact Box Formats on Comprehension of Benefits and Harms of Prostate Cancer Screening: A Randomized Trial. *Medical Decision Making*, 39(1), 41-56.)

AUTHORS' RESPONSE: As suggested, we have updated our discussion on potential methods for presenting information about benefits and risks of screening, with reference to the two recommended papers: "The best available evidence could be presented in transparent fact boxes—a tabular format that facilitates comparisons of outcomes in groups of screened vs. unscreened individuals[73]. Fact boxes can also include simple graphs such as icon arrays to allow visual comparison of quantities[74], which could be especially beneficial for people with low numeracy." (p. 19)

REVIEWER 2, COMMENT 15: suggest: "it has been argued that such claims are misleading"

AUTHORS' RESPONSE: This has been edited as suggested.

REVIEWER 2, COMMENT 16: "it is also possible that not all codes we assessed are essential for..."

AUTHORS' RESPONSE: This has been edited as suggested, and the statement has been further shortened for simplicity: "it is possible that not all codes assessed are essential for..."

REVIEWER 2, COMMENT 17: please make sure that Figure 2 is larger

AUTHORS' RESPONSE: We have enlarged Figure 2 and now use a resolution of 300 dpi, which should allow enlarging it further at the proof stage if needed.

REVIEWER 3

(Seok Won Jin, The University of Memphis, TN, USA)

REVIEWER 3, COMMENT 1: (p. 3, line 15). Design: it would clearer if you could clarify the 'information items,' given the word counts' limit.

AUTHORS' RESPONSE: To address this point, we now state: "Design: Cross-sectional analysis of websites using a comprehensive checklist containing information items on screening benefits, risks, possible results, and cervical cancer statistics."

REVIEWER 3, COMMENT 2: (p. 5, line 5-8). While most cases of cervical cancer are caused by infection with HPV, not all necessary. Please clarify the statement.

AUTHORS' RESPONSE: We have clarified that "it is caused in most cases by an infection with the human papillomavirus (HPV)"

REVIEWER 3, COMMENT 3: (p. 5, line 5-11). Appropriate in-text citations seem to be needed.

AUTHORS' RESPONSE: As requested, we have added relevant in-text citations: "Cervical cancer is highly preventable. It is caused in most cases by an infection with the human papillomavirus (HPV), which may lead to abnormal changes in cervical cells[1,2]. Such abnormalities can be detected through screening and treated before they become cancerous[3]."

REVIEWER 3, COMMENT 4: (p. 6, line 21-22). Numeracy is usually known as one of health literacy construct components so it would be clearer to say "health literacy (e.g., numeracy)"

AUTHORS' RESPONSE: This sentence has been removed in the process of shortening the paper.

REVIEWER 3, COMMENT 5: (p. 9, line 24-39) mechanic error (i.e., "cervical cancer screening"; "smear test", followed by "cervical screening"). The same on p. 14, line 5-9 and 47-48, p. 15 line 19-20, and p. 16, line 3-4.

AUTHORS' RESPONSE: We have corrected all the errors and now all punctuation marks are placed inside the quotation marks instead of outside (e.g., "cervical cancer screening," cervical smear," and "smear test.")

REVIEWER 3, COMMENT 6: (p. 9) Website selection: in Introduction, the authors indicated that organized screening programs provide websites about cervical screening (p. 5-6, line 51-6). If so, it is imperative to make sure including these websites for the analyses. Please provide the clarification on this where appropriate in this section.

AUTHORS' RESPONSE: We have clarified this: "After exclusions, 12 websites remained (Figure 1), including 4 websites from public health service providers in the UK's four countries (NHS Choices England, NHS Inform Scotland, Public Health Agency Northern Ireland, Public Health Wales) (...) All websites from public health service providers included links to official invitation leaflets, which were coded with the corresponding website, as described above." (pp. 13-14)

VERSION 2 – REVIEW

REVIEWER	Felix G. Rebitschek Max Planck Institute for Human Development, Harding Center for Risk Literacy, Germany
REVIEW RETURNED	27-Aug-2019

GENERAL COMMENTS	Thanks for providing the improved revision, which adressed my concerns! One major point left is about the abstract/interpretation/limitation and it is based on a comparison of your conclusion and your findings (Figure 2). In your results you report, for instance, that the
---

	recommendation to avoid the use of verbal quantifiers without numbers[31] was often not followed for information about risks - 3 informed with numerical information. This is not only per se against recommendations, but even more the voluntary asymmetry (at the same time numbers for benefits are presented) is problematic. You provide a lot of good evidence showing how different means are taken to increase participation. The statement in the abstract that the pages „not fully support informed decision making about screening participation“ could not be tested by your research. Informed decision-making is not signalled by mentioning the concept. Informed decision-making relies on the presence and absence of certain features in the health information on the page (e.g. absolute risk figures for both benefits and harms). Discussion on features can be found here: Bunge M, Mühlhauser I, Steckelberger A. What constitutes evidence-based patient information? Overview of discussed criteria. Patient Education and Counseling 2009;78:316—28. What I want to say: Please consider 1) pointing out in the abstract that most pages appeal to increase screening uptake [10/12 acc. to Tables S3a and b], with at least 6 mentioning the individual choice. Recommended principles of information presentation to enable informed decision-making, however, was often not followed. 2) for your limitation the whole concept of "evidence-based information" (you started with this in Table 1 actually without going back to this point in the discussion). To give another example, you code "lives saved yearly" - even if assuming that this figures could be true - there is no evidence that this is relevant information to support informed decision-making. So, necessary future studies should work with an evidence-based checklist (e.g. evidence-based guideline on presentation of health information), in the meaning of evidence for presenting/omitting certain information in a certain way – this would strengthen the appeals section. Minor points:  - version with corrections: P55/L24 -- word cancer is missing - Figure 2 is not labeled as being Figure 2 in the Figure caption - Table S1 suggest to round all traffic figures to Million that make them easily comparable - Table S2: column head "Example" is missing - perhaps refer to the discussion whether precancerous lesions are overdiagnosed; the medical community will come discussing about what are the overdiagnoses in cervical cancer - suggest that different platforms not only affect readability, but perhaps other information are presented/highlighted, as information seeking behavior (preparing screening decisions) on smartphones is substantially different Felix Rebitschek I always sign my reviews
--	--

VERSION 2 – AUTHOR RESPONSE

REVIEWER 2

(Felix G. Rebitschek, Harding Center for Risk Literacy, Max Planck Institute for Human Development)

REVIEWER COMMENT 1: One major point left is about the abstract/interpretation/limitation and it is based on a comparison of your conclusion and your findings (Figure 2). In your results you report, for instance, that the recommendation to avoid the use of verbal quantifiers without numbers[31] was often not followed for information about risks - 3 informed with numerical information. This is not only per se against recommendations, but even more the voluntary asymmetry (at the same time numbers for benefits are presented) is problematic. You provide a lot of good evidence showing how different means are taken to increase participation. The statement in the abstract that the pages „not fully support informed decision making about screening participation“ could not be tested by your research. Informed decision-making is not signalled by mentioning the concept. Informed decision-making relies on the presence and absence of certain features in the health information on the page (e.g. absolute risk figures for both benefits and harms). Discussion on features can be found here: Bunge M, Mühlhauser I, Steckelberger A. What constitutes evidence-based patient information? Overview of discussed criteria. Patient Education and Counseling 2009;78:316—28.

What I want to say:

Please consider

1) pointing out in the abstract that most pages appeal to increase screening uptake [10/12 acc. to Tables S3a and b], with at least 6 mentioning the individual choice. Recommended principles of information presentation to enable informed decision-making, however, was often not followed.

2) for your limitation the whole concept of "evidence-based information" (you started with this in Table 1 actually without going back to this point in the discussion). To give another example, you code "lives saved yearly" - even if assuming that this figures could be true - there is no evidence that this is relevant information to support informed decision-making. So, necessary future studies should work with an evidence-based checklist (e.g. evidence-based guideline on presentation of health information), in the meaning of evidence for presenting/omitting certain information in a certain way – this would strengthen the appeals section.

AUTHORS' RESPONSE: We have addressed this comment in three ways. First, we have emphasized the existing asymmetry in the presentation of benefits and risks, pointed out by the reviewer. Specifically, we now mention this in the abstract ("benefits and risks were presented using different formats, potentially hindering comparisons") and in the Discussion (p. 18): "However, we also found that websites generally presented benefits and risks using different formats (numbers vs. verbal quantifiers, respectively), potentially hindering benefit-risk comparisons."

Second, we have removed the problematic claim mentioned by the reviewer (i.e. "UK websites about cervical screening may not fully support informed decision making about screening participation"). Instead, as suggested by the reviewer (point 1 above) in the abstract we now mention explicitly that "appeals for participation were present in most websites, with almost half also mentioning informed decision making" and that "recommendations from the literature to facilitate understanding of quantitative information and facilitate informed decisions were often not followed"

Third, as suggested we now also emphasize the importance of evidence-based guidelines in relation to future research (point 2 above). In doing so we cite the paper by Bunge et al. recommended by the reviewer: "Finally, future studies should also examine how screening decisions (e.g., intentions to participate[87]) are affected by information about specific benefits and/or risks presented using different formats, building on the risk communication literature (e.g., [88]). Such work would provide valuable insights to inform the design of evidence-based public communications about screening." (p. 22)

Minor points:

REVIEWER COMMENT 2: version with corrections: P55/L24 -- word cancer is missing

AUTHORS' RESPONSE: The word 'cancer' has been added.

REVIEWER COMMENT 3: Figure 2 is not labeled as being Figure 2 in the Figure caption

AUTHORS' RESPONSE: 'Figure 2' has been added to the figure caption.

REVIEWER COMMENT 4: Table S1 suggest to round all traffic figures to Million that make them easily comparable

AUTHORS' RESPONSE: All traffic figures in Table S1 have been rounded to Million as suggested.

REVIEWER COMMENT 5: Table S2: column head "Example" is missing

AUTHORS' RESPONSE: The column head "Example statements" has been added to Table S2

REVIEWER COMMENT 6: perhaps refer to the discussion whether precancerous lesions are overdiagnosed; the medical community will come discussing about what are the overdiagnoses in cervical cancer

AUTHORS' RESPONSE: We have emphasized that overdiagnosis in cervical cancer focuses on abnormal cells. Specifically, in the Introduction (p. 5) we have clarified that risks of screening include "the detection of indolent abnormal cells that would have cleared up on their own (i.e., overdiagnosis), potentially leading to unnecessary treatment".

Similarly, in the Discussion (p. 18) we now also state explicitly that "cervical screening can lead to overdiagnosis of indolent abnormalities (potentially causing unnecessary anxiety or worries)"

REVIEWER COMMENT 7: suggest that different platforms not only affect readability, but perhaps other information are presented/highlighted, as information seeking behavior (preparing screening decisions) on smartphones is substantially different

AUTHORS' RESPONSE: We now also mention these possibilities in the Discussion (p. 21): "It would also be interesting to compare different platforms (e.g., desktop websites vs. mobile tailored versions), and to examine whether different design features (e.g., information position[80]) affect information readability, saliency, or users' information seeking behavior."

VERSION 3 - REVIEW

REVIEWER	Felix G. Rebitschek Max Planck Institute for Human Development Harding Center for Risk Literacy Germany
REVIEW RETURNED	09-Sep-2019

GENERAL COMMENTS	The authors addressed the items that I brought up.. I always sign my reviews. Felix Rebitschek
--